# Heart Failure Association-International Cardio-Oncology Society Risk Score Validation in HER2-Positive Breast Cancer

**DOI:** 10.3390/jcm12041278

**Published:** 2023-02-06

**Authors:** Michael Cronin, Aileen Crowley, Matthew G. Davey, Peter Ryan, Mahmoud Abdelshafy, Ahmed Elkoumy, Hesham Elzomor, Shahram Arsang-Jang, Sandra Ganly, Patrick Nash, James Crowley, Faisal Sharif, Andrew Simpkin, Aoife Lowery, William Wijns, Michael Kerin, Osama Soliman

**Affiliations:** 1Discipline of Cardiology, Galway University Hospital, Health Service Executive, H91 TK33 Galway, Ireland; 2CORRIB Core Lab., University of Galway, H91 TK33 Galway, Ireland; 3Discipline of Surgery, Lambe Institute for Translational Research, University of Galway, H91 TK33 Galway, Ireland; 4Precision Cardio-Oncology Research Enterprise (P-CORE), H91 TK33 Galway, Ireland; 5CURAM Centre for Medical Devices, H91 TK33 Galway, Ireland

**Keywords:** heart failure, breast cancer, HER2 therapy, anthracycline, risk prediction

## Abstract

Background: This paper looks to validate the risk score from the Heart Failure Association of the European Society of Cardiology and the International Cardio-Oncology Society (HFA-ICOS) for predicting potential cardiotoxicity from anticancer therapy for patients positive for human epidermal growth factor receptor 2. Methods: A total of 507 patients with at least five years since index diagnosis of breast cancer were retrospectively divided according to the HFA-ICOS risk proforma. According to level of risk, these groups were assessed for rates of cardiotoxicity via mixed-effect Bayesian logistic regression model. Results: A follow-up of five years observed cardiotoxicity of 3.3% (*n* = 3) in the low-risk, 3.3% (*n* = 10) in the medium-risk, 4.4% (*n* = 6) in the high-risk, and 38% (*n* = 6) in the very-high-risk groups respectively. For cardiac events related to treatment, the risk was significantly higher for the very-high-risk category of HFA-ICOS compared to other categories (Beta = 3.1, 95% CrI: 1.5, 4.8). For overall cardiotoxicity related to treatment, the area under the curve was 0.643 (CI 95%: 0.51, 0.76), with 26.1% (95% CI: 8%, 44%) sensitivity and 97.9% (95% CI: 96%, 99%) specificity. Conclusions: The HFA-ICOS risk score has moderate power in predicting cancer therapy–related cardiotoxicity in HER2-positive breast cancer patients.

## 1. Introduction

Breast cancer is one of the most common malignant cancers and has a number of different variants, including oestrogen receptor (ER) positive, progesterone receptor (PR) positive, and triple-negative cancer [1]. Approximately 20% of all breast cancers are human epidermal growth factor receptor 2 (HER2) positive, meaning that their tumours show an overexpression of HER2 or an amplification of its oncogene [2].

Trastuzumab is a humanised, recombinant monoclonal antibody that targets the extracellular domain of HER2 [3]. However, trastuzumab has been shown to cause decrease in the left ventricular systolic function [3]. Anthracyclines are a class of chemotherapeutic drugs used in HER2+ breast cancer, either in isolation or in the adjuvant setting. However, it can produce reactive oxygen species with ATP depletion and cell death, leading to anthracycline-induced cardiotoxicity [4]. This cardiotoxicity has been shown to be dose dependent, although less so with modern dosing regimens [5].

The Heart Failure Association (HFA) of the European Society of Cardiology, in conjunction with the International Cardio-Oncology Society (ICOS), published the HFA-ICOS cardiotoxic risk scoring proformas in 2020 for seven different chemotherapeutic drug classes [6]. This publication included cancer therapy–related cardiotoxicity (CTRCT) as a decrease in the left ventricular ejection fraction (LVEF) of >10% and/or to a level below 50% [6]. Previous validation has suggested that the risk score proformas outlined by HFA-ICOS underestimate the level of estimated cardiotoxic risk faced by HER2+ breast cancer patients who have received trastuzumab-based regimens [7]. This paper aims to further validate the HFA-ICOS risk score proformas in an Irish population of HER2-positive breast cancer patients.

## 2. Materials and Methods

### 2.1. Participants

Patients with HER2-positive breast cancers were recruited retrospectively from University Hospital Galway and other Saolta University healthcare group–affiliated hospitals. These same patients had been prospectively recruited into the INCEPTION study between 2005 and 2019. Inclusion criteria were as follows: ≥18 years of age, ≥5 years of follow-up, and diseases that were deemed to be curable. This study was approved by the Galway University Hospital research ethics board.

### 2.2. Data Collection

All available baseline characteristics, demographics, cardiovascular comorbidities, tumour characteristics, tumour therapy characteristics, oncologic data, and cardiologic outcome data were collected into a REDCap central database. Thiesedata contained binary and continuous variables. Binary variables included: smoking history, cause of death (cardiac reasons or not), diabetes mellitus, chronic kidney disease, hypertension, angina, cardiomyopathy, congestive heart failure, coronary artery disease, peripheral vascular disease, atrial fibrillation, and dyslipidaemia, use of medications relevant to cardiac risk (beta blockers, statins, anticoagulants, diuretics, calcium antagonists, angiotensin converting enzyme [ACE] inhibitors, antiplatelets, and angiotensin receptor blockers), previous radiotherapy, use of chemotherapy (pertuzumab, doxorubicin, epirubicin, carboplatin, docetaxel, paclitaxel), use of endocrine therapy, use of trastuzumab, and HER2 status. Continuous variables within the dataset included: date of birth, age at diagnosis, date of death, dose of radiotherapy, ER status and score, PR status and score, grade, original size, and histology. LVEF was recorded as a continuous variable available at baseline and subsequently every three months for the duration of patient trastuzumab use. A full table of binary, continuous, and outcome variables is available in the Appendix A.

Breast cancer molecular subtype was allocated based on the 12th St. Gallen expert consensus (2013) [8]. Details regarding the oncological diagnosis and plan of care of the study population are provided elsewhere [9]. There is no restriction on the availability of materials or information.

### 2.3. Risk Stratification

Patients were stratified for the risk of developing CTRCT into low-, medium-, high-, and very-high-risk groups based on the HFA-ICOS risk score proforma for HER2-targeted therapies. Each factor was classified as a medium 1, medium 2, high, or very-high-risk factor. Patients were scored and then stratified into four groups based on the number and type of risk factors that were relevant to them. The characteristics recorded above were used to score and stratify the patients based on the score assigned by the proforma. See Figure 1 illustrating this stratification.

### 2.4. Study Outcomes and Definitions

Adverse cardiac outcomes were defined as the occurrence of any or a combination of each of the following: (1) a decline in LVEF of ≥10% during or after therapy compared to baseline, (2) a decline in baseline LVEF below 50% during or after therapy, (3) the temporary or permanent discontinuation of trastuzumab therapy due to the development of cardiotoxicity, (4) the development of congestive heart failure (New York Heart Association (NYHA) class of 2–4, or 5) death due to a cardiovascular event.

### 2.5. Echocardiography

Transthoracic echocardiograms were performed using a GE ultrasound system (Vivid E9, GE Healthcare, Milmaukee, WI, USA) by certified sonographers within a European Society of Cardiology–accredited echocardiography laboratory. Comprehensive echocardiographic assessment was performed in accordance with ASE/EAE guidelines [10], including two-dimensional echocardiography and all Doppler recordings (colour, continuous, pulsed, and tissue Doppler) to assess the LV systolic (ejection fraction) and diastolic functions, and associated valvular disease. Echocardiographic analysis including chamber quantification was performed according to the ASE/EAC guideline recommendation for chamber quantification. Left ventricular end-diastolic volume and end-systolic volume were calculated from the apical four-chamber and two-chamber views using a method of discs summation, as was the left ventricular ejection fraction (by modified bi-plane Simpson rule).

### 2.6. Statistical Analysis

Patient baseline characteristics and cancer characteristics were tabulated and compared between individuals with and without cardiotoxicity events, between age groups (<65 years and ≥65 years), and between the HFA-ICOS risk groups (low, medium, high, and very high). These comparisons were made using the Bayesian Hamiltonian Monte Carlo inference. The age cut of was set at 65 and over as comorbidities and cardiac disorders are often more frequent in older patients. In addition, cardiotoxicity has been reported more often in patients older than 65 years of age. Use of anticancer treatments was compared against the age groups and HFA-ICOS risk groups.

The occurrence of CTRCT was used to calculate the actual (observed) risk faced by the patients, which was then compared to the risk estimated (predicted) by the HFA-ICOS using the logistic regression model with a Bernoulli logit link function. The Rhat, leave-one-out (LOO) cross-validation, Watanabe-Akaike information criterion (WAIC), and post-probability plots were used to check the accuracy of the regression models as well as estimated parameters. According to our results, the accuracy of models was acceptable.

To assess the adequacy and power of HFA-ICOS for prediction of CTRCT, we calculated the C-statistics, sensitivity, specificity, positive predictive value (PPV), negative predictive value (NPV), and both positive and negative likelihood ratios. The C-statistics value higher than 0.7 indicates good discriminatory power.

The associations considered statistically significant with a posterior 95% credible interval (CrI) covered 0. The rstan, brms, ggplot2, and ROCpsych packages in an R 4.1.3 environment were used to perform statistical analysis.

## 3. Results

### 3.1. Patient Characteristics

The entire cohort of 507 patients from the INCEPTION database were included, which consisted of HER2+ breast cancer patients being treated with trastuzumab and other cancer therapies including surgery, chemotherapy, and radiotherapy. Table 1 separates our cohort into those who experienced cardiotoxicity and those who did not. It lists their characteristics and tumour characteristics as well as patients’ comorbidities and concurrent cardiac medications. According to our results, the prevalence of hypertension and previous trastuzumab were higher in individuals with cardiotoxicity compared to the patients without cardiotoxicity. Unadjusted patient characteristics are available in the Appendix A.

There were no observed differences in the tumour characteristics between the patients aged over 65 and those under 65. No patient at baseline had a diagnosis of coronary artery disease. Those over the age of 65 were more likely to have hypertension, chronic kidney disease, and previous congestive heart failure, while there was no difference between the age groups in the prevalence of diabetes mellitus and cardiomyopathy. Younger patients were also more likely to smoke. A higher percentage of patients ≥65 years old were taking cardioprotective therapies at the onset of the study, with this trend seen across all the therapies with data listed.

### 3.2. Treatment Characteristics

The patients’ chemotherapeutic treatment characteristics are detailed in Table 2A (based on presence of cardiotoxicity) and Table 2B (based on HFA-ICOS risk score). More of the younger patients were receiving concurrent chemotherapy when compared to the older patients. This can be seen in all four of the concurrent treatments. A higher percentage of younger patients had received previous regimens of trastuzumab and anthracycline regimens. The taxane docetaxel was the most-prescribed chemotherapy, with 47.3% of all patients taking it. High-risk and very-high-risk patients were more likely to receive doxorubicin and paclitaxel than the medium- and low-risk patients. The reverse is true for the administration on docetaxel and carboplatin.

There was no observed difference between the uses of epirubicin amongst the two age groups. Epirubicin use was only recorded in low-risk patients. Only 4.1% of patients were administered pertuzumab, and its use did not differ significantly across the age groups or risk score groups. Of the 302 patients that tested positive for the ER+ form of breast cancer, 135 (44.7%) of them received tamoxifen therapy and 142 (47%) were prescribed the aromatase inhibitor letrozole. About 13.6% (41/302) of patients received other aromatase inhibitors. More very-high-risk patients received aromatase inhibitors. Older patients tended not to receive any endocrine therapy, nor did high-risk patients.

Radiotherapy was administered in 328 (64.7%) of the patients and was administered to the breast from which the tumour was excised. More patients <65 received radiotherapy in comparison to patients >65. The majority of the risk score groups had a similar percentage of radiotherapy use, except for the high-risk score group. The only concurrent cardiac therapy recorded was the administration of aspirin, with a higher percentage of patients >65 taking the drug at baseline.

### 3.3. CTRCT and Cardioprotective Therapies

CTRCT was recorded in 23 (4.5%) of the study patients (Table 3). Two deaths are adjudicated due to adverse cardiac events (0.4% of all patients), with a high and very high risk of cardiotoxicity based on the HFA-ICOS score. Seven (1.4%) patients had an LVEF decline of equal to or greater than 10%, with 14 (2.8%) patients having an LVEF decline below 50%. Diastolic dysfunction was included in this study as it was felt to be a clinically important heart failure syndrome that had previously been omitted within other validation studies, with four patients (0.8%) achieving this endpoint via the previously stated imaging guidelines (10).

Table 4 outlines the different cardioprotective measures taken after cardiotoxicity was detected. Four patients (0.8%) were prescribed either ACE inhibitors or angiotensin converting blockers. Five patients (1%) were administered statins. A higher percentage of patients ≥65 were prescribed these. Three patients (0.6%) received aspirin and two (0.4%) diuretics. Two patients (0.4%) received alpha blockers, four (0.8%) received beta blockers, and two (0.4%) received calcium channel blockers. No difference was seen between the age groups or risk score groups for any of these three therapies.

### 3.4. HFA-ICOS Risk Score Performance

Based on the HFA-ICOS risk stratification proforma, 100 patients (19.7%) had low risk of cardiotoxicity, 301 (59.4%) had medium risk, 90 (17.8%) had high risk, and 16 (3.2%) had very high risk of cardiotoxicity. Figure 2 demonstrates the association between cardiotoxicity and the HFA-ICOS risk score subgroups using low risk as a reference category. According to the mixed-effect Bayesian logistic regression for binary outcomes adjusted for age effect, the significantly higher risk of cardiotoxicity was estimated for the very-high-risk category of HFA-ICOS compared to the high- (Beta = 2.63, 95% CrI: 1.18, 4.18), medium- (Beta = 2.85, 95% CrI: −1.6, 4.07), and low-risk (Beta = 3.07, 95% CrI: 1.5, 4.7) categories. This same figure shows the risk differences of overall cardiotoxicity were insignificant among low-risk, medium-risk, and high-risk categories of HFA-ICOS.

Figure 3 shows the individual cardiotoxic event probability across the different cohorts with confidence intervals. In overall cardiotoxicity related to treatment, the area under the curve (AUC) was 0.643 (CI 95%: 0.51, 0.76), with 26.1% (95% CI: 8%, 44%) sensitivity and 97.9% (95% CI: 96%, 99%) specificity (see Table 5). In addition, the highest AUC was estimated as a 10% decline in LVEF event (Table 5). The full results of prediction ability of HFA-ICOS are shown in Table 5, with the highest AUC estimated for death followed by LVEF decline ≥10%. The results of post-power analysis for each estimated C-statistics are included. Receiver operating characteristic curves for cardiotoxicity are included in the Appendix A. To check the power of our results we performed post-power analysis, with the estimated powers for any cardiotoxic event and LVEF decline >10%, 69.3%, and 68.6% respectively. In addition, estimated power for predictability of diastolic dysfunction was 9.1%.

Increased risk of cardiotoxicity based on the HFA-ICOS risk category correlated to the increasing cardiac events observed in trastuzumab use, with the overall levels of observed cardiotoxicity reaching 3 (3%) in low-risk patients, 10 (3.3%) in medium-risk, 4 (4.4%) in high-risk, and 6 (38%) in very-high-risk patients (Table 3). This compared to a predicted cardiotoxicity risk by the HFA-ICOS of <2%, 2–9%, 10–19%, and >20% risk. As regards cardiac events related to cancer treatment, the HFA-ICOS risk score model carried a sensitivity of 0.26 and a specificity of 0.98 (Table 5). The proforma also had a PPV of 0.375 and an NPV 0.965 in predicting cardiac events related to cancer treatment.

## 4. Discussion

We do not have sufficient evidence to reject the hypothesis that the HFA-ICOS risk proforma underestimates the cardiotoxic risk faced by HER2+ breast cancer patients receiving trastuzumab. Indeed, we can summarize our findings by stating that the risk score has moderate performance in this cohort and could not be used effectively across all the different cohorts to predict patients who will have cardiotoxic events. Overall, the score performed well in predicting cardiotoxicity in very-high-risk patients and had excellent negative predictive power.

Amongst the four groups, the two groups that fell within the estimations of the HFA-ICOS were those in the medium-risk category and very-high-risk category. Therefore, it can be concluded that the risk score system does not have a strong predictive power. However, the majority of patients were in the medium-risk cohort and the rate of occurrence was correct in this group, and this is a positive finding. The HFA-ICOS risk scoring proformas were able to significantly stratify the patients amongst the risk groups, with each higher group receiving a greater risk of cardiotoxicity. This is in keeping with previous validation [7].

The HFA and ICOS have recommended seven different risk scoring proformas, each dedicated to patients undergoing therapy with an individual chemotherapeutic drug class. These classes are anthracyclines, HER2 targeted therapies (including trastuzumab), vascular endothelial growth factor inhibitors, proteasome inhibitors and immunomodulatory drugs, RAF and MEK inhibitors used in combination, multi-targeted kinase inhibitors, and androgen deprivation therapies. Immune checkpoint inhibitors were mentioned in the paper but did not receive their own risk scoring proforma. Each risk score proforma consists of several factors related to the patient, and those related to cancer therapy. The sum score is calculated based on the cumulative risk derived from the presence of one or more risk factors. However, these estimations, just like the risk factors, are only based on expert opinion and have not yet been validated in prospective studies, and are largely derived from non-randomized studies as abovementioned.

As mentioned, trastuzumab and anthracyclines, such as doxorubicin, are often used in conjunction with one another, usually with trastuzumab acting as an adjuvant therapy after the initial cancer treatment. Together these have shown good efficacy in preventing breast cancer recurrence [11]. However, studies have shown that the combination of the two drugs increases a patient’s risk of cardiotoxicity, with a five-year combined heart failure and cardiomyopathy rate of 20.1% for patients receiving both, as recorded by the Cancer Research Network [12]. An alternative regimen that has been shown to be nearly as effective as anthracycline and trastuzumab adjuvant therapies is a combination of docetaxel (a taxane), carboplatin (a platinum metal), and trastuzumab. This treatment, as mentioned, is nearly as efficacious as anthracycline–trastuzumab therapies but also a lot safer, showing far less cardiotoxicity and should undergo further study [13].

Battisti et al. published the first, and currently only, paper to attempt to validate the risk score proformas proposed by the HFA and ICOS. They specifically attempted to validate the risk proforma for HER2-targeted therapies in an English population of 931 breast cancer patients [7]. In their paper they demonstrated patients at low-risk of cardiotoxicity had a 7-fold higher risk than expected, yet this paper concluded that the HFA-ICOS proformas accurately predicted cardiotoxicity. Thus, further validation attempts are needed to see if these results can be replicated and to prove the actual predictive power of the HFA-ICOS risk proformas.

A potential factor affecting the HFA-ICOS risk scoring proformas is the inaccurate scoring of certain risk factors. Examples include diabetes mellitus and chronic kidney disease. The HFA-ICOS risk score proformas assign these variables a risk score of medium 1. However, diabetes has been associated with right ventricular dysfunction and structural ventricular remodeling independent of other factors [14]. Chronic kidney disease is associated with coronary microvascular disease, which is a precursor to left ventricular dysfunction [15].

Our data showed the increased prevalence of hypertension and use of trastuzumab in those experiencing cardiotoxicity. Hypertension is ascribed medium 1, and previous trastuzumab use (without cardiotoxicity) is not included in the score. Therefore, future risk scoring methods should assign a higher risk score to these factors, and include previous trastuzumab use, as they pose a greater risk to patients than assigned by the HFA-ICOS proformas.

It has been shown in previous studies that older patients with cardiac comorbidities have a higher risk of developing trastuzumab-induced cardiotoxicity [16]. However, interestingly, there was no significant difference seen between the level of cardiac events between the older and younger age groups. This could be due to the significantly higher levels of anthracycline administration and radiotherapy seen in the younger population. Radiotherapy has been linked with the development of coronary artery disease, which is often a precursor to myocardial infarction. Exposure of the left side of the chest to radiation, which occurred as both breasts were exposed to radiation in these patients, has been linked to congestive heart failure [17]. Anthracycline cardiotoxicity has been detailed above, however the damage caused by anthracyclines to a younger heart leave it more vulnerable to the development of future cardiomyopathies, with minor insults that would not affect a normal healthy heart resulting in ischemic damage to the affected heart [18].

These factors are likely the reason why the level of adverse cardiac events is closely balanced across the age groups. Further study needs to investigate other possible risk factors for chemotherapy-induced cardiotoxicity, such as genetic factors [19]. These risk factors, along with the changes regarding hypertension and trastuzumab use we have proposed, should be included in any future risk scoring methods. The risk factors should be heavily researched for their connection to cardiotoxicity and accurately scored. These risk scores should be validated in the study in which they are outlined.

### Limitations

This is a retrospective, single-centre analysis which suffers from the typical limitations of the retrospective study design. This includes missing or incomplete data for some of the variables, which may lead to incorrect patients’ stratification. Data on variables such as patient body mass index, ECG features, and Charleson comorbidity index was missing or incomplete and thus the patients could not be assessed under these factors. Global longitudinal strain was not routinely performed. All 507 patients were reviewed by two faculty cardiologists and two cardiology senior fellows, who recorded any documentation of depressed LVEF or cardiotoxicity by attending internal medical physicians, medical oncologists, or cardiologists. We consider all 507 patients to have been included in this study appropriately. Our centre does not have access to all the DICOM files for echocardiograms, as it operates as a quaternary referral hospital and facilitates a shared care program with satellite hospitals, who perform routine surveillance for cardiotoxicity within these patients via local cardiac diagnostic services. However, the 507 patients were all closely monitored as they received their anticancer treatment in the quaternary centre. In this context we have 100% of the data regarding our stated endpoints of cardiotoxicity, which were all identified via the medical file review.

## 5. Conclusions

The current HFA-ICOS risk scoring method has moderate power to predict the risk of adverse cardiac events in our population of HER2+ breast cancer patients. Estimation of overall risk was correct in the medium- and very-high-risk cohort, however for low- and high-risk cohorts, cardiotoxicity remained outside of the predicted rate of occurrence. The majority of the patients were in the medium-risk cohort and the rate of occurrence was correct in this group. The low-risk cohort appear to have similar outcomes. Lastly, the score has a strong negative predictive value. We would encourage a higher score to hypertension than medium 1, and the inclusion of previous trastuzumab use that did not result in cardiotoxicity within the risk score. Given this, we would suggest its continued use with alterations to the scoring system as described. Future study would be needed to validate these changes.

## Figures and Tables

**Figure 1 jcm-12-01278-f001:**
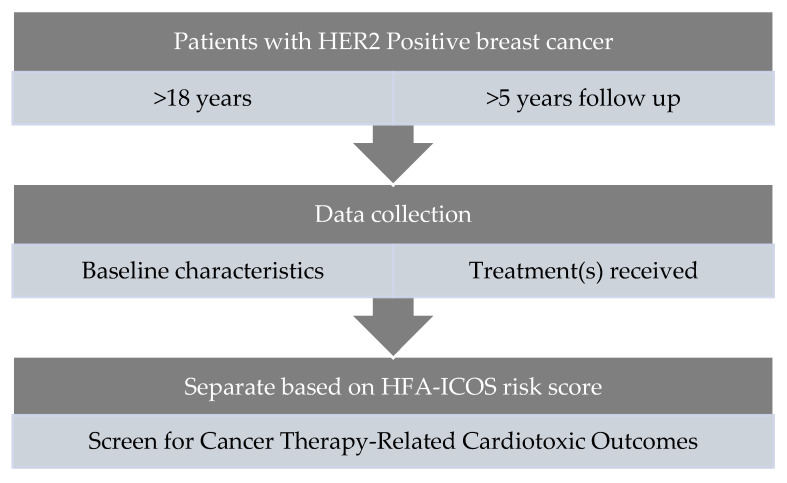
Data Collection Methods.

**Figure 2 jcm-12-01278-f002:**
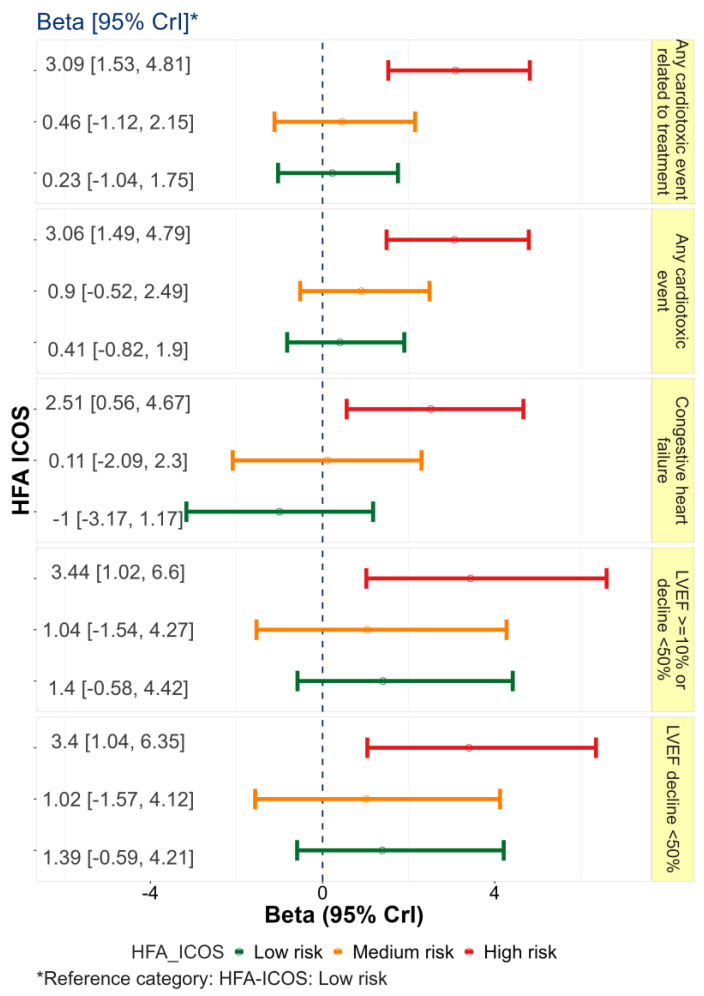
Association between cardiotoxicity and the HFA−ICOS risk score subgroups using low risk as a reference category.

**Figure 3 jcm-12-01278-f003:**
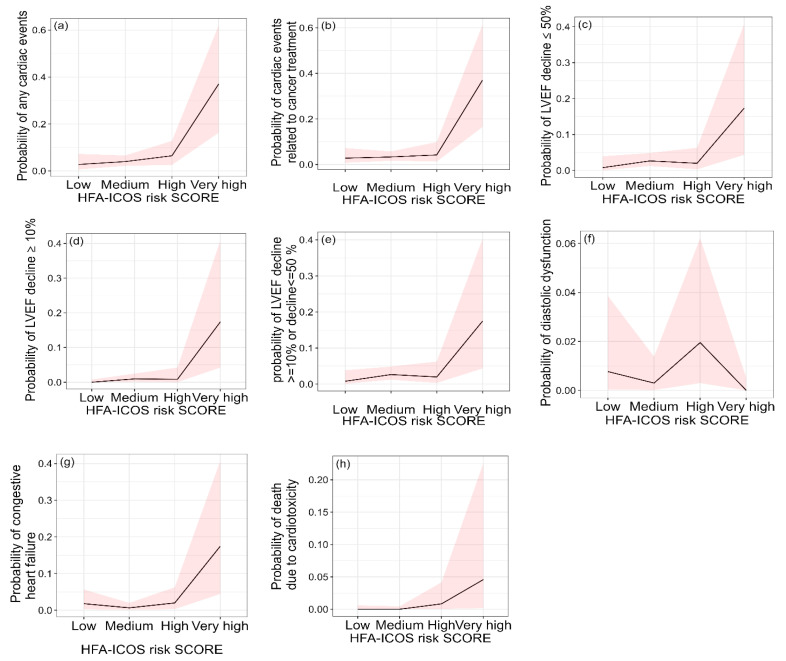
The probability of cardiotoxicity related to treatment across HFA-ICOS score risk categories. The subfigures (**a**–**h**) are split into various cardiotoxic outcomes and confidence intervals are provided for cardiotoxic event probability in each subgroup.

**Table 1 jcm-12-01278-t001:** Baseline characteristics in populations by cardiotoxicity.

	With Cardiotoxicity N (%)	Without Cardiotoxicity N (%)	Posterior Beta	95% Credible Interval for Beta
Characteristics				
Age (>65 year/<65 year)	5/18 (21.7)	128/356 (26.4)	−0.319	(−1.42, 0.63)
Diabetes	1/22 (4.3)	22/440 (4.8)	−0.624	(−3.8, 1.38)
Hypertension	10/13 (43.5)	96/366 (20.8)	1.055	(0.171, 1.9)
Kidney disease	3/20 (13)	28/434 (6.1)	0.69	(−0.83, 1.92)
Congestive cardiac failure	5 (1.3)	12 (9)		
Smoking	5/16 (23.8)	101/331 (23.4)	−0.04	(−1.19, 0.94)
Death	17/6 (73.9)	253/230 (52.4)	0.991	(0.05, 2.01)
Tumour Features				
Grade 0 or 1	0 (5.3)	31 (8.3)	-	-
Grade 2	7 (36.9)	176 (33.8)	51.87	(1.55, 215.14)
Grade 3	16 (57.8)	277 (57.9)	52.28	(1.99, 215.6)
Previous trastuzumab	22/1 (95.7)	353/131 (72.9)	2.5	(0.62, 5.44)
ER status (positive/negative)	14/9 (59.4)	288/193 (60.2)	−0.06	(−0.94, 0.79)
PgR status (positive/negative)	11/12 (47.8)	229/251 (47.7)	0.003	(−0.84, 0.85)
Histology				
Ductal	313 (83.7)	108 (81.2)	0.45	(−0.73, 1.94)
Lobular	1/22 (4.3)	23/461 (4.8)	−0.62	(−3.7, 1.36)
Mixed	0/23 (0)	4/480 (0.8)	-	-
Other	2/21 (8.7)	56/428 (11.6)	−0.541	(−2.46, 0.85)
Cardiac Therapy				
Aspirin	0	3/49 (0.6)	-	-
Beta blockers	0/23 (0)	53/431 (0.8)	-	-
Statins	37 (9.9)	48 (36.1)	0.001	
Anticoagulant	23/0 (100)	432/48 (89.3)	28.4	(1.78, 118.88)
ACE inhibitor	0	4/48 (12)	-	-

**Table 2 jcm-12-01278-t002:** (**A**): Previous and concurrent breast cancer therapy characteristics by age. (**B**): Breast cancer therapy characteristics by HFA-ICOS risk score.

(A)					
**Treatment Characteristic**	**Category**	**With Cardiotoxicity N (%)**	**Without Cardiotoxicity N (%)**	**Posterior Beta**	**95% Credible Interval for Beta**
Concurrent chemotherapy	Paclitaxel; (yes/no)	6/15 (28.6%)	92/318 (22.4%)	0.273	(−0.8, 1.22)
	Docetaxel; (yes/no)	14/7 (66.7%)	226/168 (57.4%)	0.41	(−0.5, 1.35)
	Anthracycline (Doxorubicin); (yes/no)	0/23 (17.2%)	39/445 (8.1%)	0.86	(−0.05, 1.73)
	Carboplatin; (yes/no)	7/14 (33.3%)	160/233 (40.7%)	−0.432	(−3.6, 1.6)
Epirubicin	-	2 (0.5%)	0 (0 %)	-	-
Pertuzumab (yes/no)	-	1/22 (4.1%)	20/464 (4.3%)	−0.08	(−0.98, 1.99)
Radiotherapy (yes/no)	-	15/6 (65.2%)	313/104 (64.7%)	−0.12	(−1.08, 0.95)
Endocrine therapy	Tamoxifen (yes/no)	8/15 (34.8%)	127/357 (26.2%)	0.376	(−0.51, 1.25)
	Letrozole (yes/no)	10/13 (43.5%)	132/352 (27.3%)	0.704	(−0.16, 1.55)
	Other Aromatase inhibitors (yes/no)	1/22 (4.3%)	40/444 (8.3%)	−1.2	(−4.23, 0.73)
**(B)**					
**Treatment Characteristic**	**Category**	**Low Risk (N = 100)**	**Medium Risk (N = 301)**	**High Risk (N = 90)**	**Very High Risk (N = 16)**
Concurrent chemotherapy	Paclitaxel; (yes/no)	13 (13%)	59 (19.6%)	20 (22.2%)	6 (37.5%)
	Docetaxel; (yes/no)	55 (55%)	146 (48.5%)	34 (37.8%)	5 (31.3%)
	Anthracycline (Doxorubicin); (yes/no)	12 (13%)	51 (17%)	19 (21.1%)	5 (31.3%)
	Carboplatin; (yes/no)	36 (36%)	108 (35.9%)	18 (20%)	5 (31.3%)
Epirubicin use	Yes	2 (2%)	0 (0%)	0 (0%)	0 (0%)
	No	78 (78%)	260 (86.4%)	78 (86.7%)	14 (87.5%)
Pertuzumab use	Yes	3 (3%)	17 (5.6%)	1 (1.1%)	0 (0%)
	No	97 (97%)	284 (94.4%)	89 (98.9%)	16 (100%)
Radiotherapy use	Yes	60 (60%)	199 (66.1%)	57 (63.3%)	12 (75%)
	No	22 (22%)	64 (21.3%)	22 (24.4%)	2 (12.5%)
	Unknown	18 (18%)	38 (12.6%)	11 (12.2%)	2 (12.5%)
Endocrine use	None	25 (25%)	113 (37.5%)	42 (46.7%)	5 (31.3%)
	Tamoxifen	31 (31%)	80 (26.6%)	22 (24.4%)	2 (12.5%)
	Letrozole	30 (30%)	83 (27.6%)	23 (25.6%)	6 (37.5%)
	Other aromatase inhibitors	9 (9%)	21 (7%)	9 (10%)	2 (12.5%)

**Table 3 jcm-12-01278-t003:** CTRCT experienced during/after treatment and during follow-up (at five years).

Cardiac Events	Overall (N = 507)	Age < 65 Years (N = 374)	Age ≥ 65 Years (N = 133)	Beta (95% CrI)	Low Risk (N = 100)	Medium Risk (N = 301)	High Risk (N = 90)	Very High Risk (N = 16)
Overall: N (%)	27 (5.3%)	19 (3.7%)	8 (1.6%)	−0.13 (−0.97, 0.78)	3 (3%)	12 (3.98%)	6 (4.5%)	6 (37.5%)
Not related to cancer treatment: N (%)	4 (0.8%)	1 (0.2%)	3 (0.6%)	-	0	2 (0.7%)	2 (0.7%)	0
Related to cancer treatment: N (%)	23 (4.5%)	18 (3.6%)	5 (1%)	0.33 (−0.63, 1.47)	3 (3%)	10 (3.3%)	4 (4.4%)	6 (37.5%)
LVEF decline ≥10%: N (%)	7 (1.4%)	6 (1.2%)	1 (0.2%)	1.16 (−0.87, 4.11)	0	3 (1%)	1 (1.1%)	3 (18.8%)
LVEF decline below 50%: N (%)	14 (2.8%)	13 (2.6%)	1 (0.2%)	1.97 (0.08, 4.85)	1 (1%)	8 (2.7%)	2 (2.2%)	3 (18.8%)
LVEF decline ≥10%/below 50%: N (%)	14 (2.8%)	13 (2.6%)	1 (0.2%)	1.94 (0.03, 4.89)	1 (1%)	8 (2.7%)	2 (2.2%)	3 (18.8%)
Congestive heart failure: N (%)	9 (1.8%)	5 (1%)	4 (0.8%)	−0.77 (−2.12, 0.64)	2 (2%)	2 (0.7%)	2 (2.2%)	3 (18.8%)
Diastolic dysfunction: N (%)	4 (0.8%)	2 (0.4%)	2 (0.4%)	0.44 (−1.83, 3.45)	1 (1%)	1 (0.7%)	2 (2.2%)	0
Death due to cardiotoxicity: N (%)	2 (0.4%)	2 (0.4%)	0 (0%)	−0.77 (−3.82, 2.43)	0	0	1 (1.1%)	1 (6.3%)

**Table 4 jcm-12-01278-t004:** Cardioprotective therapies administered.

Treatment Characteristic Category	Overall (N = 507)	Age < 65 Years (N = 374)	Age ≥ 65 Years (N = 133)	*p*-Value	Low Risk (N = 100)	Medium Risk (N = 301)	High Risk (N = 90)	Very High Risk (N = 16)	*p*-Value
(ACEi/ARB)	4 (0.8%)	1 (0.3%)	3 (2.3%)	0.002	0 (0%)	4 (1.3%)	0 (0%)	0 (0%)	0.12
Alpha blocker	2 (0.4%)	1 (0.3%)	1 (0.8%)	0.070	0 (0%)	2 (0.7%)	0 (0%)	0 (0%)	0.13
Beta blocker	4 (0.8%)	3 (0.8%)	1 (0.8%)	0.19	2 (2%)	2 (0.7%)	0 (0%)	0 (0%)	0.078
Statins	5 (1%)	1 (0.3%)	4 (3%)	0.001	0 (0%)	5 (1.7%)	0 (0%)	0 (0%)	0.13
CCB	2 (0.4%)	1 (0.3%)	1 (0.8%)	0.085	0 (0%)	2 (0.7%)	0 (0%)	0 (0%)	0.32
Aspirin	3 (0.6%)	1 (0.3%)	2 (1.5%)	0.001	0 (0%)	2 (0.7%)	1 (1.1%)	0 (0%)	0.20
Diuretic	2 (0.4%)	0 (0%)	2 (1.5%)	0.007	0 (0%)	1 (0.3%)	1 (1.1%)	0 (0%)	0.079

**Table 5 jcm-12-01278-t005:** Statistical analysis of cardiotoxic outcomes.

Cardiac Events	Sensitivity	Specificity	AUC, 95% CI	PPV	NPV	Accuracy
Overall	0.26	0.97	0.652 (0.53, 0.76)	0.113	0.963	0.785
Related to cancer treatment	0.26	0.98	0.643 (0.51, 0.76)	0.375	0.965	0.947
LVEF decline ≥10%	0.42	0.974	0.762 (0.55, 0.97)	0.188	0.992	0.966
LVEF decline below 50%	0.214	0.973	0.629 (0.47, 0.78)	0.188	0.978	0.953
LVEF decline ≥10%/below 50%	0.214	0.974	0.629 (0.47, 0.78)	0.188	0.978	0.953
CHF	0.55	0.79	0.658 (0.46, 0.85)	0.047	0.99	0.793
Diastolic dysfunction	0.50	0.79	0.588 (0.29, 0.88)	0.019	0.995	0.791

## Data Availability

All data is available upon request.

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
