# Peer review of "Heart Failure Association-International Cardio-Oncology Society Risk Score Validation in HER2-Positive Breast Cancer"

_jcm, 2023, doi:10.3390/jcm12041278_

Round 1
Reviewer 1 Report
Overall an excellent article trying to use easy tools to risk stratify breast cancer patients and cardiac toxicity. Minor changes suggested below prior to publication.
1. Methods - a table for binary and continuous variables may be helpful as it was a lot of wording in that paragraph and could simplify for the reader.
2. Methods - Lots of tables that were overwhelming at times. Try to summarize them more or use more selective ones to convey your point.
3. Discussion - work is needed on grammatical errors and sentence flow structure. It will help it read better and flow to the reader.
4. Conclusion - how will your research results affect daily clinical practice or what would you change based on your results? Should everyone use this scale? Should we not offer certain breast cancer treatments for higher risk patients, administer lower doses, or just monitor those high risk ones more closely?
Author Response
We would like to thank the Editors and the Reviewers for their insightful comments. We appreciated all feedback that was provided, and as a group reflected on our research. Following a careful revision process we now strongly believe that our manuscript has meet the recommendations for publication by Peer Review. Please find below a point-by-point reply to each of the reviewer’s comments. They are highlighted within the revised text by Microsoft Word “Track Changes” function.
Comment 1: “Methods - a table for binary and continuous variables may be helpful as it was a lot of wording in that paragraph and could simplify for the reader.”
Answer 1: We want to thank the reviewer for her/his suggestion. A table in this regard has been fashioned and can be found as supplementary material.
Comment 2: “Methods - Lots of tables that were overwhelming at times. Try to summarize them more or use more selective ones to convey your point.”
Answer 2: We want to thank the reviewer for her/his suggestion. We have increased space around the tables in the text and formatted the title to aid ease of consumption. We have reviewed the text and make reference to significant findings in the results and conclusion section.
Comment 3: “Discussion - work is needed on grammatical errors and sentence flow structure. It will help it read better and flow to the reader.”
Answer 3: We want to thank the reviewer for her/his suggestion. Grammatical errors have been amended (see tracked changes) with some fresh paragraph spacing to aid ease of reading. On page 11, two paragraphs have been switched in order to provide an improved flow of information in an easier to follow fashion.
Comment 4: “Conclusion - how will your research results affect daily clinical practice or what would you change based on your results? Should everyone use this scale? Should we not offer certain breast cancer treatments for higher risk patients, administer lower doses, or just monitor those high risk ones more closely?”
Answer 4: We want to thank the reviewer for her/his suggestion. For clarity of conclusion we have added into conclusion statement that we would suggest use of scale with changes to its scoring system as described in the text. As an authorship group we feel that until we know the score is as good as it could be that it should be used in isolation to alter clinical practice.
Reviewer 2 Report
I suggest authors to add whether ACEi/ARBs or ARNI were prescribed or why not? Especially, considering their cardiac protection.
Please, add whether pts had IHD or non-IHD diagnosis prior to the cancer treatment or none of them were diagnosed with heart disease?
Author Response
We would like to thank the Editors and the Reviewers for their insightful comments. We appreciated all feedback that was provided, and as a group reflected on our research. Following a careful revision process we now strongly believe that our manuscript has meet the recommendations for publication by Peer Review. Please find below a point-by-point reply to each of the reviewer’s comments. They are highlighted within the revised text by Microsoft Word “Track Changes” function.
Comment 1: “I suggest authors to add whether ACEi/ARBs or ARNI were prescribed or why not? Especially, considering their cardiac protection.
Answer 1: We want to thank the reviewer for her/his suggestion. From review of our dataset we are aware of the patients who are prescribed ACEi/ARB therapy (see table 4 in manuscript file). From those who are not prescribed the medications we have not recorded the indication. From the 27 patients that had a cancer-therapy related cardiotoxic outcome, we know 4 patients had diastolic failure, 3 had chronic kidney disease and 2 had an outcome of death. Diastolic failure in isolation may not have provoked the primary physician to prescribe ACEi/ARB (we are unsure if hypertension was the cause of all cases of diastolic dysfunction). Further the severity of the 3 patients with chronic kidney disease may have precluded safe ACEi/ARB therapy. Lastly 2 with an outcome of death may have suffered this suddenly. Beyond these 9 patients we remain uncertain, and unfortunately as to why 23 patients were not prescribed ACE/ARB/ARNi therapy remains speculation.
Comment 2: “Please, add whether pts had IHD or non-IHD diagnosis prior to the cancer treatment or none of them were diagnosed with heart disease?”
Answer 2: We want to thank the reviewer for her/his suggestion. No patient had this diagnosis at baseline and an addition has been made to the text in this regard on page 5 in the manuscript.
Reviewer 3 Report
Manuscript jcm-2137142
In this research article entitled “Heart Failure Association‑International Cardio‑Oncology Society Risk Score Validation in HER2‑positive Breast Cancer”, the authors study the potential cardiotoxicity from anti-cancer therapy for patients positive for human epidermal growth factor receptor 2 (HER2), particularly, the HFA-ICOS risk score in an Irish population of HER2 positive breast cancer patients.
Hereafter, some points that should be taken into account before final publication.
Comments to the authors:
- The manuscript lacks mechanistic illustrations. In fact, there are several included mechanisms within the article but authors used just text and no figures at all. I think it would be better to include some illustrations, as sometimes it’s more informative to the readers.
- The readers would certainly like to see some additional characteristics in the studied populations, as some are relevant to the concerned study as charleson index and electrocardiogram data.
- Figure 1 is of low quality regardless of the small font size. Authors have to ameliorate it.
- The English language is fine, just small editing is required.
Author Response
We would like to thank the Editors and the Reviewers for their insightful comments. We appreciated all feedback that was provided, and as a group reflected on our research. Following a careful revision process we now strongly believe that our manuscript has meet the recommendations for publication by Peer Review. Please find below a point-by-point reply to each of the reviewer’s comments. They are highlighted within the revised text by Microsoft Word “Track Changes” function.
Comment 1: “The manuscript lacks mechanistic illustrations. In fact, there are several included mechanisms within the article but authors used just text and no figures at all. I think it would be better to include some illustrations, as sometimes it’s more informative to the readers.”
Answer 1: We want to thank the reviewer for her/his suggestion. As recommended we have created an illustration regarding our methods which can be found on page 3.
Comment 2: “The readers would certainly like to see some additional characteristics in the studied populations, as some are relevant to the concerned study as charleson index and electrocardiogram data.”
Answer 2: We want to thank the reviewer for her/his suggestion. Having reviewed the dataset, unfortunately we do not have specific information regarding Charleson Index and ECG features. This is regrettable and has been listed as a limitation of the paper in the limitations section on page 12.
Comment 3: ”Figure 1 is of low quality regardless of the small font size. Authors have to ameliorate it”
Answer 3: We want to thank the reviewer for her/his suggestion. This has been addressed and the ameliorated image has replaced the original in the text on page 9.
Comment 4: “The English language is fine, just small editing is required.”
Answer 4: We want to thank the reviewer for her/his suggestion. Please find grammatical editing as suggested throughout document via Microsoft “Track Changes” function.
Round 2
Reviewer 3 Report
After providing a revised version of the manuscript, it is obviously noticed that the manuscript has been ameliorated following the comments, which have been addressed by the reviewers. The revised version of the manuscript warrant publication in JCM.